# Multi-organ segmentation based on 2.5D semi-supervised

Hao Chen[1][0000−0002−6849−9413], Wen Zhang[1][0000−0001−8679−5462],
Xiaochao Yan[1][0000−0002−4358−6588], Yanbin Chen[1][0000−0003−0231−1390],
Xin Chen[1][0000−0003−3039−4939], Mengjun Wu[1][0000−0002−6176−4560],
Lin Pan[1][0000−0002−6176−4560], and Shaohua Zheng[1(✉)][0000−0002−9133−705X]

Intelligent Image Processing and Analysis Laboratory, Fuzhou University, Fuzhou FJ
350108, CHN `sunphen@fzu.edu.cn`

**Abstract.** Automatic segmentation of multiple organs is a challenging topic. Most existing approaches are based on 2D network or 3D network, which leads to insufficient contextual exploration in organ segmentation. In recent years, many methods for automatic segmentation based on fully supervised deep learning have been proposed. However, it is very expensive and time-consuming for experienced medical practitioners to annotate a large number of pixels. In this paper, we propose a new two-dimensional multi slices semi-supervised method to perform the task of abdominal organ segmentation. The network adopts the information along the z-axis direction in CT images, preserves and exploits the useful temporal information in adjacent slices. Besides, we combine Cross-Entropy Loss and Dice Loss as loss functions to improve the performance of our method. We apply a teacher-student model with Exponential Moving Average(EMA) strategy to leverage the unlabeled data. The student model is trained with labeled data, and the teacher model is obtained by smoothing the student model weights via EMA. The pseudo-labels of unlabeled images predicted by the teacher model are used to train the student model as the final model. The mean DSC for all cases we obtained on the validation set was 0.5684, the mean NSD was 0.5971, and the total run time was 783.14s.

**Keywords:** Semi-supervised · Deep learning · Organ segmentation.

## 1 Introduction

Automatic segmentation of multiple organs is a challenging topic. The main problems in medical segmentation can be outlined as follows: (1) Manual annotation of organs from CT scans is time-consuming and laborious. (2) Medical data involves patient privacy issues. In recent years, many proposed fully supervised deep learning automatic segmentation methods rely on a large number of pixel-level annotations, but the annotation of multiple organs is very expensive and time-consuming. In addition, existing 2D methods cannot fully utilize spatial information, and 3D methods consume a lot of computational resources. This paper

proposes a novel 2.5D multi-slice semi-supervised approach to perform abdominal organ segmentation. Considering the limited labeled data, semi-supervised learning is applied to explore useful information from unlabeled data.

Integrated learning and other power-hungry algorithms can achieve wonderful segmentation results. However, it will eventually lead to a very bloated model. In contrast, the lightweight model needs low requirements on device. It is easier to be deployed in real-world applications. The main theme of the Fast and Low-resource semi-supervised Abdominal Organ Segmentation in CT 2022(FLARE2022) Challenge is to propose a solution with high efficiency and high accuracy as the benchmark while occupying low GPU and CPU resources. It is important to have both lightweight model and low resource usage, as well as high accuracy and efficiency. A representative of the current state-of-the-art approaches is nnU-Net [5], which provides a fully automated end-to-end segmentation method and comes out top in several competitions. The model has good performance on segmentation, but it is also not light enough and requires a lot of GPU memory.

In this paper, our contributions are listed as follows:

1. We propose a 2.5D semi-supervised multi-organ segmentation framework. It introduces connected adjacent slices as input [7] to improve the utilization of 3D information with only a few increase in computational resources. It employs a teacher-student semi-supervised strategy to use unlabeled data.

2. We use 2D U-Net [5] as the main network framework. The attention module of Convolutional Block Attention Module (CBAM) [14] is added to improve the data information mining.

3. EMA [6] is applied to optimize the teacher model, which makes the performance of model more robust.

## 2   Method

### 2.1   Preprocessing

**Threshold truncation** According to our observation of CT files, the contrast between organ tissues is more obvious when the threshold is taken near [-250,300], so we truncate the CT threshold on this basis.

**Cropping strategy** In the training phase, the images and labels are cropped according to the slices containing labels and the slices without the labels are discarded. In the inference phase, the area to be segmented in the CT image is concentrated in the middle of z-axis. The larger the image size is, the more unrelated areas exist in the CT. So we use center cropping to cut the number of z-axis slices to the nearest power 2. For example, a CT image size is 809×512×512(Z×H×W), we can cut it to 512×512×512 to speed up the subsequent data reading.

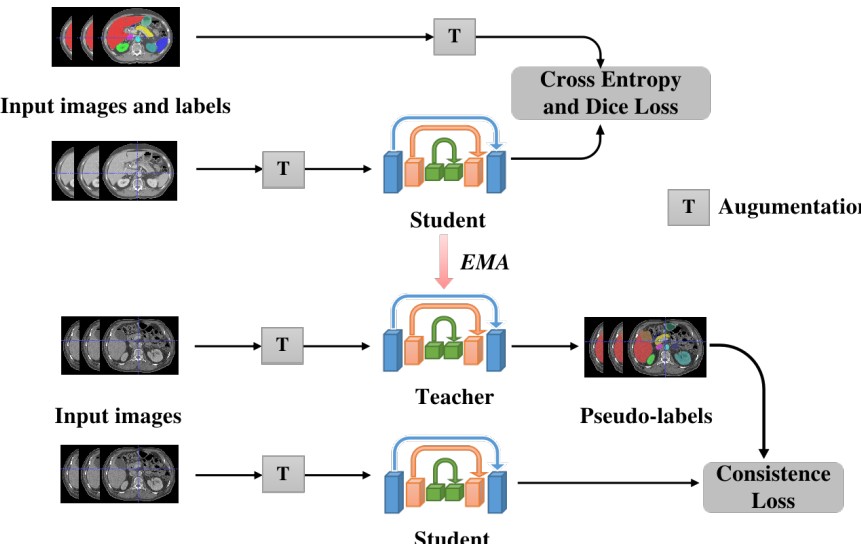

**Fig. 1.** The workflow of our method. The labeled data are fed into the student network after data augmentation. The student network is passed through EMA to obtain a more robust teacher model. The unlabeled data are pseudo-labeled by the teacher network after data augmentation. The pseudo-labels are sent to the student network with the corresponding unlabeled data.

**Resampling method for anisotropic data** We adjust the slice size during data loading to reduce GPU memory usage. After testing with different sizes, the 2D slice size of $128\times128$ has an impact on model performance, while $256\times256$ takes up more memory without much performance improvement. So we resample the 2D slice from the original $512\times512$ to a size of $192\times192$. Meanwhile, during the scaling process, we use trilinear sampling to ensure that the variation of image texture features is as small as possible. We use nearest neighbor sampling on labels to ensure that the label values remain unchanged.

**Intensity normalization method** To make the data more easily compute and improve the performance of model, we normalize and standardize the threshold value to [0, 1].

### 2.2 Proposed Method

Figure 1 illustrates the general workflow of our semi-supervised segmentation network.

Firstly, we train the student model on labeled data.

Secondly, we use the EMA method to obtain teacher model by smoothing the student model weights. The loss curve of the model using EMA is smoother

and has less jitter on the image [6]. The model does not fluctuate significantly due to some abnormal loss values. So the robustness of the model after EMA optimization is better.

Thirdly, the teacher model predicts the unlabeled data to generate pseudo-labels. Then we continue to train the student model on the unlabeled data with pseudo-labels. Finally, we get the final student model.

The teacher-student model uses the U-Net-CBAM network architecture. Figure 2 illustrates the Network architecture of our method.

We apply the attention mechanism to the U-Net segmentation network. The CBAM allows better focus on prominent areas and suppress irrelevant background regions. It can be well embedded in the CNN framework. Compared with other attention modules, the model performance can be improved without adding too much computational effort [14].

**Number setting of adjacent slices** We investigated some papers [13]. It is best to use 3 adjacent slices, while making full use of context information. In terms of details, 3 adjacent slices are connected on the channel, fed into the model for training and inference. It makes the best possible use of 3D information while slightly increasing the GPU memory footprint.

**Relevant principles of EMA** In depth learning, the weight of the model will shake at the actual best point in the last n steps of each training, and there will be an abnormal value relative to the best point. Therefore, we take the average of the last n steps to make the model more robust. Since the value of n is a decreasing process, it is equivalent to a sliding average process.

**Strategies to use the unlabelled cases** We use the teacher model to predict unlabeled data to get pseudo-labels and use consistency loss to allow student models to learn the content of unlabeled data. We believe that too much unlabeled data will magnify the error information in the pseudo-labels and bring greater impact on the model. Therefore, only part of the unlabeled data is used for training. We randomly selected 100 data before the start of the training.

**Network architecture details description** In Figure 2, our proposed U-Net-CBAM network consists of a combination of U-Net network and CBAM. CBAM includes the spatial attention module and the channel attention module. The network input first goes through 5 convolution modules and 4 max pooling layers to complete the downsampling process, and then passes through 5 convolution modules and four upsampling to get the output. The output of each layer of the downsampling path is connected by the features of the skip connection and the upsampling path, respectively. The skip connection performs channel-wise and spatial-wise feature correction via the CBAM.

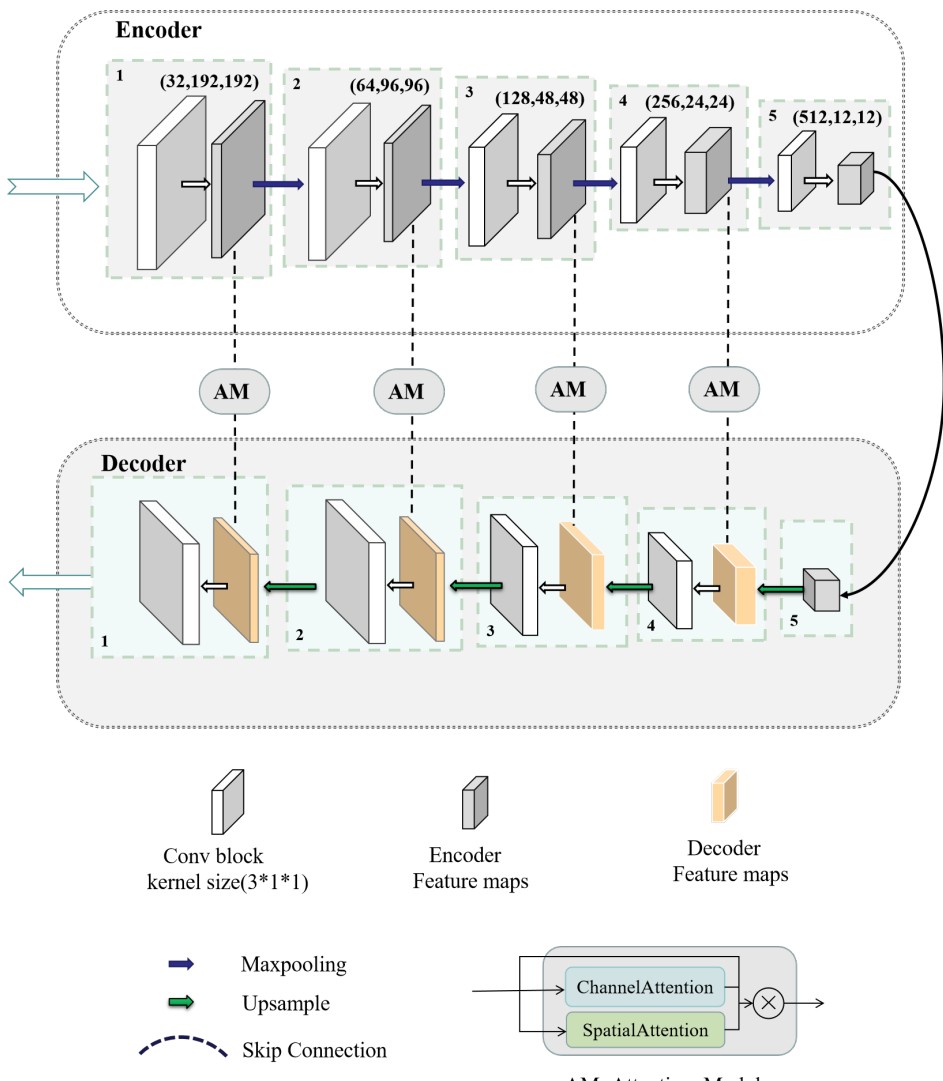

**Fig. 2.** Network architecture. U-Net-CBAM is based on U-Net with additional CBAM in the jump connection can better extract the image features and details.

**Loss function [8]** We set up two loss functions on our method. They are Supervised loss function and Unsupervised loss function.

1) Supervised loss function: It consists of Cross-Entropy Loss($L_{ce}$) [2] and Dice Loss($L_{dice}$) [11] with weights, and the percentages of both are $\alpha$ and $1-\alpha$, respectively. The $L_{dice}$ comes with weights to facilitate the adjustment of the percentages between different categories.

$$Loss = \alpha * L_{ce}(g,p) + (1-\alpha) * L_{dice}(g,p) \tag{1}$$

where $g$ is ground truth, $p$ is prediction, we use $\alpha$ to balance the $L_{ce}$ and $L_{dice}$. $\alpha$ is a constant weight to balance the $L_{ce}$ and $L_{dice}$. Here we set it to 0.5.

$$L_{dice} = 1 - \sum_{i=0}^{k-1} \beta_i * Dice(g_i, p_i) \tag{2}$$

where $g_i$ is ground truth, $p_i$ is prediction, $\beta_i$ represents weight of every class. The ratio between any two categories is set to 1:1.

$$Dice(g,p) = \frac{2|g \cap p|}{|g| + |p|} \tag{3}$$

where $g$ and $p$ is what we want to compare by using dice.

$$L_{ce}(g,p) = -\sum_{i=1}^{C} g_i \log(p_i) \tag{4}$$

where $C$ denotes the number of classes, $g_i$ is ground truth, $p_i$ is prediction.

2) Unsupervised loss function: we use consistency loss($L_{csst}$) [6], which directly measures the consistency between prediction and pseudo-label.

$$L_{csst} = \|M_s(A_1(x)) - M_t(A_2(x))\|_2^2 \tag{5}$$

where $x$ is input image, $A_1(\cdot)$ and $A_2(\cdot)$ are different noise addition functions, $M_s(\cdot)$ and $M_t(\cdot)$ denotes output of model of student and teacher.

**Strategies to improve inference speed and reduce consumption** Firstly, we use U-Net network architecture and combine it with CBAM. Compared with other attention modules, it has less parameters and consumes less computational resources.

Secondly, we use multiple slices to avoid losing 3D information. Besides, we resize the image as shown in the preprocessing stage to reduce the GPU memory consumption without affecting the model performance.

### 2.3   Post-processing

Operation 1: We perform a median filtering operation with a convolution kernel of 5×5×5 on the image. It can optimize the edges of the segmentation results.

Operation 2: We resize the image in preprocessing. After obtaining the segmentation results, we scale the predicted values to the original size in post-processing based on the nearest neighbor interpolation operation.

When we perform operation 1, some of the segmentation results disappear, even if reducing the size of the convolution kernel. We finally abandon operation 1.

## 3 Experiments

### 3.1 Dataset and evaluation measures

The FLARE 2022 is an extension of the FLARE 2021 [9] with more segmentation targets and more diverse abdomen CT scans. The dataset is curated from more than 20 medical groups under the license permission, including MSD [12], KiTS [3,4], AbdomenCT-1K [10], and TCIA [1]. The training set includes 50 labelled CT scans with pancreas disease and 2000 unlabelled CT scans with liver, kidney, spleen, or pancreas diseases. The validation set includes 50 CT scans with liver, kidney, spleen, or pancreas diseases. The testing set includes 200 CT scans where 100 cases have liver, kidney, spleen, or pancreas diseases and the other 100 cases have uterine corpus endometrial, urothelial bladder, stomach, sarcomas, or ovarian diseases. All the CT scans only have image information, while the center information is not available.

The evaluation measures consist of two accuracy measures: Dice Similarity Coefficient (DSC) and Normalized Surface Dice (NSD), and three running efficiency measures: running time, area under GPU memory-time curve, and area under CPU utilization-time curve. All measures will be used to compute the ranking. Moreover, the GPU memory consumption has a 2 GB tolerance.

### 3.2 Implementation details

**Environment settings** We performed the training and inference process based on the environment of Table 1.

**Table 1.** Development environments and requirements.

| | |
|---|---|
| Windows/Ubuntu version | Ubuntu 18.04.6 LTS |
| CPU | Intel(R) Core(TM) i5-7500 CPU@3.40GHz |
| RAM | 4×4GB; 2400MT/s |
| GPU (number and type) | 1 NVIDIA RTX 2080 (8G) |
| CUDA version | 11.4 |
| Programming language | Python 3.8 |
| Deep learning framework | Pytorch (Torch 1.7.0, torchvision 0.8.2) |
| Specific dependencies | medicaltorch, pandas, scipy, collections |
| (Optional) Link to code | |

**Training protocols** In training the teacher and student models, we optimized the models using the Adam optimizer with an initial learning rate of 0.001 and a learning rate reduction strategy using CosineAnnealingLR. The training protocols are presented in Table 2 and Table 3.

**Data augmentation** In training teacher and student model, we used elastic transformation, random horizontal and vertical flipping and random rotation for data enhancement.

The elastic transformation is to generate a random standard deviation for each dimension of the pixel in the (-1,1) interval. It filters the deviation matrix of each dimension with a Gaussian filter $(0, \sigma)$. The final amplification factor $\alpha$ is used to control the deviation range. We set $\sigma$ to fluctuate between (10.0, 20.0) and $\alpha$ to fluctuate between (2.0, 4.0). We set the random possibility to 0.3.

Random horizontal and vertical flipping is to randomly rotate the image vertically and horizontally. We set the random possibility to 0.5.

Random Rotation is to randomly select an angle (0, 90, 180, 270) to rotate the image.

**Table 2.** Training protocols.

| | |
|---|---|
| Network initialization | Teacher Net |
| Batch size | 16 |
| Patch size | $3 \times 192 \times 192$ |
| Total epochs | 150 |
| Optimizer | Adam |
| Initial learning rate (lr) | 0.001 |
| Lr decay schedule | CosineAnnealingLR |
| Training time | 3.25 hours |
| Number of model parameters | 74.1M[1] |
| Number of flops | 8.22G[2] |
| $CO_2$eq | 1 Kg[3] |
| Loss function | Cross-Entropy Loss and Dice Loss[4] |

## 4    Results and discussion

### 4.1    Quantitative results on validation set

On the provided validation dataset, we perform ablation experiments and the results are shown in Table 4. The purpose is to compare the model performance on the validation set using only labeled data and using both labeled and unlabeled data. When using only labeled dataset, the DSC reaches 0.5454, while when using both labeled and unlabeled dataset, the DSC reaches 0.5684.

**Table 3.** Training protocols for the refine model.

| | |
|---|---|
| Network initialization | Student Net |
| Batch size | 8 |
| Patch size | 3×192×192 |
| Total epochs | 100 |
| Optimizer | Adam |
| Initial learning rate (lr) | 0.001 |
| Lr decay schedule | CosineAnnealingLR |
| Training time | 16 hours |
| Number of model parameters | 74.1M[5] |
| Number of flops | 8.65G[6] |
| $CO_2$eq | 1 Kg[7] |
| Loss function | Consistence Loss[8] |

After using unlabeled data, the segmentation results improve, and the previously under-segmented organs can be initially segmented. The improvement in the DSC illustrates that using pseudo-labels to exploit unlabeled data can improve model performance.

In this paper, the performance of the proposed method is evaluated using the provided validation set with ground truth. The evaluation metrics are DSC and NSD. The results are shown in Table 4. The average value of DSC is 0.5684 and the average value of NSD is 0.5971.

We can find that the model performs better in the segmentation of three organs, the Liver, Spleen, and Aorta, but is slightly weak in the other organs.

### 4.2    Qualitative results on validation set

The segmentation excellence results are shown in the Figure 3 and the results of the poor segmentation are shown in the Figure 4.

Our method makes excellent segmentation results on case #08 and #21, but makes poorer segmentation results on case #30 and #48. The supervised method on case #08, #21, #30, #48 have over-segmentation results than the semi-supervised method.

We demonstrate ablation experiments in a visual manner. The experiment results verity the effectiveness of semi-supervised method. They are all from the provided validation set with ground truth.

Possible reasons for the failure of cases or organ segmentation are listed as follows:

1) We use fewer adjacent slices for training and inference.

2) Some of the organ segments are lost in the preprocessing because some information is lost in the scaling of the images.

3) The pseudo-labels from the teacher model may contain error information, which the student model may have learned in training.

**Table 4.** Overview of DSC and NSD metrics on validation set.

| Name | Mean(DSC) | Mean(DSC) | Mean(NSD) | Mean(NSD) |
|------|-----------|-----------|-----------|-----------|
| Metohd | w/o ssl | w ssl | w/o ssl | w ssl |
| Liver | 0.8079 | 0.8496 | 0.7320 | 0.7718 |
| RK | 0.4590 | 0.4816 | 0.3773 | 0.4022 |
| Spleen | 0.7169 | 0.7073 | 0.6400 | 0.6475 |
| Pancreas | 0.5177 | 0.5463 | 0.6109 | 0.6447 |
| Aorta | 0.8525 | 0.8703 | 0.8578 | 0.8811 |
| IVC | 0.6561 | 0.6671 | 0.6219 | 0.6424 |
| RAG | 0.4057 | 0.3968 | 0.5415 | 0.5458 |
| LAG | 0.3045 | 0.3750 | 0.4236 | 0.5130 |
| Gallbladder | 0.2460 | 0.3540 | 0.1918 | 0.3026 |
| Esophagus | 0.5981 | 0.5638 | 0.7202 | 0.6917 |
| Stomach | 0.5650 | 0.5890 | 0.5580 | 0.5736 |
| Duodenum | 0.3685 | 0.3957 | 0.5475 | 0.5911 |
| LK | 0.5916 | 0.5931 | 0.5307 | 0.5544 |
| Average | 0.5454 | 0.5684 | 0.5656 | 0.5971 |

### 4.3   Segmentation efficiency results on validation set

The segmentation efficiency results of the validation set are shown in Table 5. It mainly includes various metrics such as CPU, GPU and runtime. The GPU-Memory usage is 1455MB, which is smaller than 2048MB. The model has a good level in Time, AUC-GPU-Time, and AUC-CPU-Time metrics, and the average values are 15.66s, 17140.12, and 348.15 respectively. We can see that the model fully meets the minimum requirements of the Challenge in terms of resource usage and consumption.

**Table 5.** Overview of segmentation efficiency on validation set.

|  | Time/s | GPU-Memory/MB | AUC-GPU-Time | AUC-CPU-Time |
|------|--------|---------------|--------------|--------------|
| Lowest | 8.53 | 1455 | 94932 | 1661.84 |
| Average | 15.66 | 1455 | 17140.12 | 348.15 |
| Highest | 74.56 | 1455 | 7644 | 181.96 |

### 4.4   Results on final testing set

Our final results on the test set are shown in Table 6. The final average DSC value is 0.6153, and the average NSD value is 0.6484. We can find that the

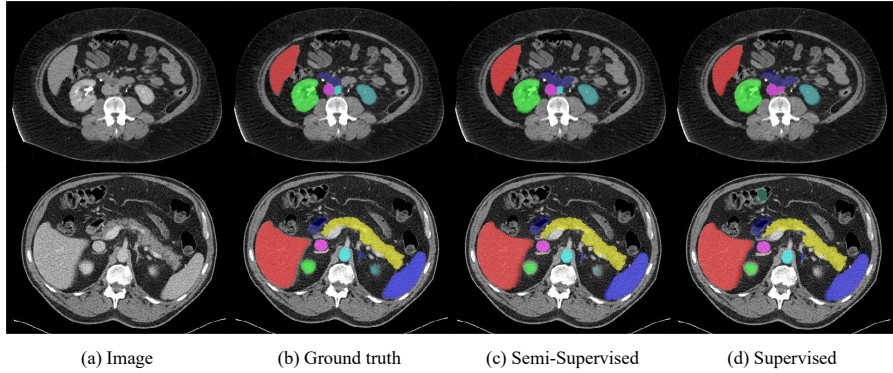

(a) Image          (b) Ground truth          (c) Semi-Supervised          (d) Supervised

**Fig. 3.** Excellent segmentation results on case #08(up) and #21(down)

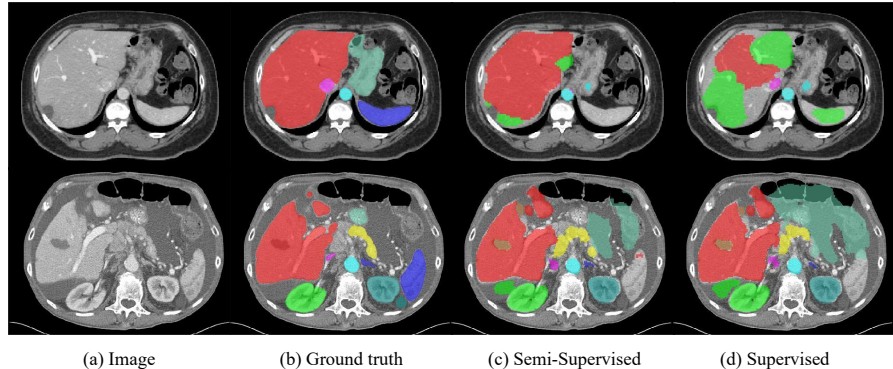

(a) Image          (b) Ground truth          (c) Semi-Supervised          (d) Supervised

**Fig. 4.** Poorer segmentation results on case #30(up) and #48(down)

model performs better in the segmentation of three organs, the Liver, Spleen, and Aorta, but is slightly weak in the other organs. The results on the test set are consistent with those of the validation set.

## 4.5   Limitation and future work

The limitation of our method is whether we can obtain good teacher models. If the teacher model cannot be well pseudo-labeled for unlabeled data, it will affect the results of semi-supervised training. This eventually leads to poor segmentation. In future work, we can improve the effect of pseudo-labeling by optimizing our method, such as improving the network architecture, post-processing the pseudo-labeling, etc.

**Table 6.** Overview of DSC and NSD metrics on test set.

| Name | Mean(DSC) | Mean(NSD) |
|------|-----------|-----------|
| Liver | 0.8723 | 0.8086 |
| RK | 0.4932 | 0.3722 |
| Spleen | 0.7976 | 0.7418 |
| Pancreas | 0.5579 | 0.6685 |
| Aorta | 0.8312 | 0.8506 |
| IVC | 0.7037 | 0.6813 |
| RAG | 0.4885 | 0.6723 |
| LAG | 0.4295 | 0.5873 |
| Gallbladder | 0.5232 | 0.4530 |
| Esophagus | 0.6180 | 0.7568 |
| Stomach | 0.6626 | 0.6699 |
| Duodenum | 0.3868 | 0.5915 |
| LK | 0.6350 | 0.5753 |
| Average | 0.6153 | 0.6484 |

## 5  Conclusion

In the work of abdominal multi-organ segmentation, we propose a 2.5D-based semi-supervised segmentation method to achieve effective use of unlabeled data and reduce the occupation of computing resources.

**Findings** The main findings are listed as follows:

1) The performance of the model is improved after using pseudo-labels to exploit unlabelled data.

2) In some cases, it is able to segment each organ very well. However, in some other cases, the segmentation of the corresponding organs is poor. After observation, we found that in these cases, there was more noise in the CT images.

3) The quality of the pseudo-label depends on the supervised training phase.

**Results** The main results are listed as follows:

1) We believe that only a part of the data distribution of some cases is learned during training. The generalization performance of the model is not very good.

2) The quality of the CT image files themselves is also important, i.e. the equipment used to take the CT images should also be checked.

3) Our teacher-student model strategy is feasible, it improves the performance of our model to some extent.

**Acknowledgements** The authors of this paper declare that the segmentation method they implemented for participation in the FLARE 2022 challenge has not used any pre-trained models nor additional datasets other than those provided

by the organizers. The proposed solution is fully automatic without any manual intervention.

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
