# OpenReview forum: "Multi-organ segmentation based on 2.5D semi-supervised"
_MICCAI.org/2022/Challenge/FLARE_

### Official Review · Reviewer_kVxr · 2022-09-12
**This paper has sufficient sections and provides clear idea despite confusing sentences, typos and lacking of in-depth analysis for the result**

**Rating:** 5
**Confidence:** 4

**Review:**

Their main contribution is stacking 3 consecutive slices on z-axis into a single 3D input for 2D U-Net with CBAM module integrated. They also use teacher-student setting with aditional consistency loss to learn unlabeled CT data.

This paper has sufficient sections and provides clear idea despite some lacking in in-depth analysis for the result. For example, in table 4 some classes like spleen, pancrease, esophagus witness decrease in Mean DSC after using semi-supervised setting.

The reviewer does not think this work has great significance in both method and result.

Some suggested improvement:
- On page 4, section .2, what is "sliding average"? what is an "abnormal value" ?
- On page 3, in the cropping strategy, what is the “take the number of slices down to the second power of 2” ? Don’t you just need to sample randomly 3 consecutive slices from the original CT volume, why need cropping on the z-axis .
- How do you do inference ? How to sample input for inference

---

> ### Author Response · Authors · 2022-10-12
> **Thank you for pointing out the problem!**
>
> 1. In depth learning, the weight of the model will shake at the actual best point in the last n steps of each training, and there will be an abnormal value (the abnormal value is relative to the best point). Therefore, we take the average of the last n steps to make the model more robust. Since the value of n is a decreasing process, it is equivalent to a sliding average process.
> 2. Our original intention here is to cut the number of z-axis slices to the nearest power 2. It is a method to speed up data loading. For example, a CT image with the size of (809, 512, 512), we can cut it to (512, 512, 512). It has nothing to do with the process of sending adjacent slices to the model.
> 3.  We send three adjacent slices(3, 192,192) on the z-axis into the model for inference. Finally we get the final result, then resize to (3, 512, 512) and conduct post-processing.   Observing the z-axis of the data, we can know that the area to be segmented in the CT image is concentrated in the middle, and the larger the image size is, the more unrelated areas exist in the CT. We cut the image to the power of 2. For example, a CT image with the size of (809, 512, 512), we can cut it to (512, 512, 512). Then we scale the xy axis to obtain(512, 192, 192)CT images. It can greatly reduces the time for data loading.

---

### Official Review · Reviewer_ZSyv · 2022-09-14
**A 2.5D semi-surpervised segmentation method**

**Rating:** 6
**Confidence:** 3

**Review:**

Summary:

In this paper, the authors propose a new two-dimensional multi slices semi supervised method to perform the task of abdominal organ segmentation. The information along the Z-axis direction in CT images utilized to preserves the useful temporal information in adjacent slices

Strengths:

A 2.5D semi supervised multi organ segmentation framework, which takes adjacent slices as input.
The EMA is used to assist the teacher model in generating pseudo labels using unlabeled data.

Weakness:

The so-called adjacent slices are not well represented in the methods section.
No corresponding studies have been conducted to reduce the computational consumption.

Details:

1. The English of your manuscript must be improved before resubmission. Many sentences contain grammatical and/or spelling mistakes or are not complete sentences. Besides, please standardize the abbreviation format of some variables, especially the case of letters.
2. The characters in Figure 1 and Figure 2 are very small, please enlarge them appropriately.
3. In Table 4, the decrease in the metrics of some organs occurred after the use of additional semi-supervised modules, and it is recommended that the authors appropriately discuss the reasons for the occurrence of this phenomenon.
4. For Section 2, is the "student model" here the same thing as the "student model" above? (i.e., is the training of "student model" mentioned here continuing after the model obtained from "student model" above or starting from scratch?) Please clarify appropriately.
5. For Section 2.1 Considering that some of the unlabeled data are used in the training process (some of the data contain a very large scan), how are these data clipped in this paper?

---

> ### Author Response · Authors · 2022-10-12
> **Thank you for pointing out the problems !**
>
> 1. We will pay attention to the modification of grammar and expression.
> 2. We will make them bigger enough.
> 3. We have proposed the possible reasons for the decrease of some organ segmentation metrics due to semi-supervision In 4.2.
> 4. Our original intention is to continue the training from the student model after the full supervision training.
> 5. Observing the z-axis of the data, we can know that the area to be segmented in the CT image is concentrated in the middle, and the larger the image size is, the more unrelated areas exist in the CT. We cut the image to the power of 2. For example, a CT image with the size of (809, 512, 512), we can cut it to (512, 512, 512). Then we scale the xy axis to obtain(512, 192, 192)CT images. It can greatly reduces the time for data loading.

---

### Official Review · Reviewer_hZ6i · 2022-09-16
**Multi-organ segmentation based on 2.5D semi-supervised**

**Rating:** 4
**Confidence:** 3

**Review:**

Strengths: The authors propose a two-dimensional multi slices semi supervised method, which avoids complete loss of 3D information and reduces memory consumption.
Weaknesses:
- The proposed method uses multi slices to perform CT segmentation. How to determine the number of slices?
- The detail training protocols are not clear. The description of parameter alpha in eq. (1) is not clear. Is alpha a constant or changing with iterations?
- During inference, how to process the test data into multi slices? How to obtain the final results based on the inference results of multi slices?

---

> ### Author Response · Authors · 2022-10-12
> **Thanks for pointing out our problems!**
>
> 1 .We investigated some papers and combined our own computer configuration.  It is the best choice to take 3 adjacent slices, while making full use of context information.
>
> 2. We will add this. Alpha is a constant weight to balance the cross entropy loss and dice loss. Here we set it to 0.5.
>
> 3. Observing the z-axis of the data, we can know that the area to be segmented in the CT image is concentrated in the middle, and the larger the image size is, the more unrelated areas exist in the CT. We cut the image to the power of 2. For example, a CT image with the size of (809, 512, 512), we can cut it to (512, 512, 512). Then we scale the xy axis to obtain(512, 192, 192)CT images. It can greatly reduces the time for data loading.  We send three adjacent slices(3, 192,192) on the z-axis into the model for inference. Finally we get the final result, then resize to (3, 512, 512) and conduct post-processing.

---

### Official Review · Reviewer_ktqp · 2022-09-18
**The authors use EMA (Exponential Moving Average) algorithm based on the teacher-student model to perform the semi-supervised segmentation task of FLARE2022. A  "2.5D" segmentation framework is used to decrease the computing resources.**

**Rating:** 6
**Confidence:** 3

**Review:**

1, The authors claim that "The network utilizes the information along the Z-axis direction in CT images and preserves and exploits the useful temporal information in adjacent slices".
The authors input the network with three slices close to one slice, e.g., 3×192×192, which come from the same axis, which I think is still a 2D network structure. What we generally think of as 2.5D is the use of cross-sampling, such as the existence of a certain angle between the three slices. This 2.5D sampling method is to avoid using dense 3D data, but also to ensure a certain amount of 3D information.

2.Very long sentences should be cut into short sentences to convey a clearer meaning. For example, the sentence in the abstract.
"In using unlabeled images, we use the EMA (Exponential Moving Average) algorithm to optimize the teacher model weights based on the teacher-student model in the process of training the student model with labeled data, allowing the teacher model to set pseudo-labels for unlabeled data and then learned by the students, using a consistency loss function to promote refinement of model performance for the purpose of using unlabeled images."
I prefer to "We use a teacher-student model with EMA (Exponential Moving Average) strategy to leverage the unlabeled data.
The student model is trained with labeled data, and the teacher model is obtained by smoothing the student model weights via EMA.
The pseudo-labels of unlabeled images, which are predicted by the teacher model, is used to train the student model as the final model."

3. The authors claim that  "In this stage, the teacher-student model uses the U-Net-CBAM network architecture, and one case of every 35 unlabeled data is selected for training." I am confused of this sentence. What is the meaning of "case" here? In section 3.1, the word "case" means an image of a patient. The authors should clarify this or replace a new word for more clear illustration.

4. At least on ablation study should performed.

5. A brief introduction to the excellent and poor results should be given. For example, As shown in Figure 3, the supervised method on case #** have over-segmentation results than the semi-supvervised method.

---

> ### Author Response · Authors · 2022-10-12
> **Thanks you for pointing out our problems!**
>
> 1. But In my opinion, 2.5D method is a method between 2D and 3D method. It means that 2.5D methods has the characteristics of both 2D and 3D methods. We use a stack of adjacent slices as input to the network .It retains the breadth information of 2D method and makes full use of 3D spatial information. So I think our method is a 2.5D method. In many papers, authors also claims that “ use a stack of adjacent slices as input to the network “ is a 2.5D methods.
> 2. Your opinion is very good. We will pay attention to the modification of grammar and expression.
> 3. Our original intention is to take part of unlabeled samples for semi supervised training.
> 4. But We have made a ablation study in Table 4.
> 5. We will briefly introduce the good and bad results at 4.2.

---

### Meta-Review · Program_Chairs · 2022-09-28

**Recommendation:** Major Revision
**Confidence:** 5

**Metareview:**

Reviewers raise many concerns and suggestions. Please address all comments in the revised manuscript.

---

> ### Author Response · Authors · 2022-10-13
> **We have made a point-by-point response to the reviewers' comments and revised our manuscript.**
>
> We have made a point-by-point response to the reviewers' comments and revised our manuscript.